# Investigation of Impact of C/Si Ratio on the Friction and Wear Behavior of Si/SiC Coatings Prepared on C/C-SiC Composites by Slurry Reaction Sintering and Chemical Vapor Infiltration



Daming Zhao [1,2], Kaifeng Cheng [2,3], Baiyang Chen [1], Peihu Gao [1,*], Qiaoqin Guo [1,*], Hao Cheng [2], Anton Naumov [4], Qiao Li [2] and Wenjie Kang [2]

1   School of Materials and Chemical Engineering, Xi'an Technological University, Xi'an 710021, China; zhaodaming@xacmkj.com (D.Z.); baiyang2578@163.com (B.C.)
2   Xi'an Chaoma Technology Co., Ltd., Xi'an 710025, China; chengkaifeng@stu.xjtu.edu.cn (K.C.); chenghao@xacmkj.com (H.C.); liqiao950129@163.com (Q.L.); kangwenjie@xacmkj.com (W.K.)
3   School of Materials Science and Engineering, Xi'an Jiaotong University, Xi'an 710079, China
4   Lightweight Materials and Structures Laboratory, Institute of Mechanical Engineering, Materials and Transport, Peter the Great St. Petersburg Polytechnic University, St. Petersburg 195251, Russia; anton.naumov@spbstu.ru
*   Correspondence: tigergaopei@163.com (P.G.); guoqiaoqin66@126.com (Q.G.); Tel.: +86-29-83208080 (P.G.); +86-29-86173324 (Q.G.)

**Abstract:** Carbon/carbon (C/C)-SiC composite materials have a series of outstanding advantages, such as a light weight, resistance to thermal degradation, excellent friction performance, and good stability in complex environments. In order to improve the wear resistance of the C/C-SiC composite matrix, Si/SiC coatings were prepared by a combination of chemical vapor infiltration and reactive sintering. The wear performance of Si/SiC coatings with different amounts of silicon carbide was investigated. When the carbon silicon ratio in the slurry was 1:3, the SiC particle content in the coating was 93.0 wt.%; the prepared Si/SiC coating exhibited the lowest wear rate of $3.2 \times 10^{-3}$ mg·N$^{-1}$·m$^{-1}$ among the four coatings; and its frictional coefficient was 0.95, which was higher than that of the substrate. As the residual Si content in the coating decreased, the continuity between SiC particles in the coating was improved. Both the high hardness of SiC and the dense coating contributed significantly to enhancing the coating's wear resistance.

**Keywords:** C/C-SiC composite materials; coating; wear resistance





## 1. Introduction

Compared with traditional brake materials (such as carbon/carbon (C/C) composite materials, metallurgical brake materials, polymer brake materials, etc.), C/C-SiC composite materials have many advantages, such as a low density (about $2.0 \pm 0.2$ g/cm$^3$), high frictional coefficient, good corrosion resistance, and long service life [1–4]. In particular, their heat resistance is many times higher than that of ordinary brake discs. Carbon/ceramic materials can be used under high-temperature conditions of 1600 °C and can effectively and stably resist thermal degradation [5,6]. However, ordinary gray cast iron brake materials experience thermal decay due to high heat during full force braking, resulting in reduced braking effectiveness [7,8]. C/C-SiC composite materials are gradually being used in luxury cars, such as Ferrari and Porsche 911 [9,10]. With the improvements in production technology and the decrease in manufacturing costs, C/C-SiC brake materials are expected to be more widely used in new energy vehicles, armored vehicles, high-speed trains, and other fields in the future [9,11,12].

C/C-SiC composite materials are often fabricated by the alternate stacking of a non-woven carbon fabric and short fiber mesh layer [2]. The non-woven fabric region contains a large number of continuous fibers, with high content of the C phase. The short fiber mesh

layer is in the ceramic phase region [13]. During the braking process, the contact area can be a random combination of these two areas [11,13,14]. However, due to the high hardness of SiC and Si as compared to the carbon phase, the softer C phase [15,16] is more prone to wear compared to the SiC and Si phases [17,18]. The friction between different areas during the braking process may generate instabilities [14,19,20]. The microstructural design of C/C-SiC for wear has attracted more attention in recent years [12–19,21]. Research shows that the wear rate of C/C-SiC brake materials increases exponentially with the braking speed [22,23]. Meanwhile, exposed carbon fibers are prone to oxidization at high temperatures, which can reduce the braking efficiency of composite materials under high-temperature conditions [24–26].

In order to improve the braking performance of C/C-SiC composite materials, a large amount of research work has been conducted. Wear-resistant and antioxidant coatings are considered one of the most effective ways to reduce the wear rate and improve their high-temperature stability performance [27]. Silicon carbide is often used to improve the high-temperature oxidation resistance, friction, and wear performance of C/C composites due to its high hardness, high strength, high-temperature chemical stability, and good chemical compatibility with the carbon matrix [28–31]. Chen et al. reported the deposition of SiC coatings on C/C composites using the chemical vapor deposition (CVD) method and studied the frictional behavior of the SiC coating [17]. The results showed that under all experimental loads, a layer of silica film was formed between the 10 and 20 N contact surfaces for SiC/SiC coupling, which was beneficial in improving the tribological properties. However, in previous reports, it was shown that SiC coatings might contain a small amount of residual Si, which could improve the toughness of SiC coatings. The influence of different Si/SiC content on the friction and wear performance of coatings appears unclear, and there are few reports on the influence of the residual Si content on the friction and wear mechanism.

In this work, Si/SiC coatings were prepared on the surfaces of C/C-SiC composite materials through slurry reaction sintering and chemical vapor filtration. The phase composition, microstructure, and friction and wear behavior of coatings with different Si/SiC components were investigated. The influence of residual silicon on the friction and wear behavior of coatings was discussed.

## 2. Material and Methods

### 2.1. Material Preparation

Si/SiC coatings were prepared on C/C-SiC composite materials in three steps. Firstly, 2.5D fabric preforms were prepared using carbon fibers with needle punching, and the 2.5D fabric preforms were densified using chemical vapor infiltration (CVI). In the CVI process, propylene was used as the precursor at a temperature of $1000 \pm 50$ °C, and the densification process lasted for $400 \pm 50$ h. After densification, the density of the porous C/C composite material was $1.4 \pm 0.1$ g/cm$^3$. The as-received porous C/C composites were machined to a 20 mm diameter and 8 mm thickness. In the second step, the SiC coating was fabricated by chemical vapor infiltration and reactive sintering. The specific method utilized different proportions of graphite powder and silicon powder mixed with phenolic resin and alcohol for an in situ reaction in a silicon atmosphere and a high-temperature furnace, as shown in Figure 1. The alcohol content was 60%, and the phenolic resin content was 10%. The weight ratio of graphite to silicon powder was controlled at 1:1, 1:2, 1:3 and 1:4, respectively. After 1 h of mechanical stirring, the slurry was scraped evenly on the surface of the received circular sample. The third step was to infiltrate the porous C/C composite material with a coating on the surface using molten silicon at a temperature of $1500 \pm 100$ °C in a silicon atmosphere for 1.5 h. The molten silicon reacted with the pyrolytic carbon (PyC) in the porous C/C composites and carbon powder in the slurry. C/C-SiC composites with Si/SiC coatings with different SiC content could be formed. The density of the C/C-SiC composite material was $2.0 \pm 0.20$ g/cm$^3$. The process is described briefly in the schematic diagram

shown in Figure 2. The densities of the samples in this work were compared with the results reported in the literature, as shown in Table 1.

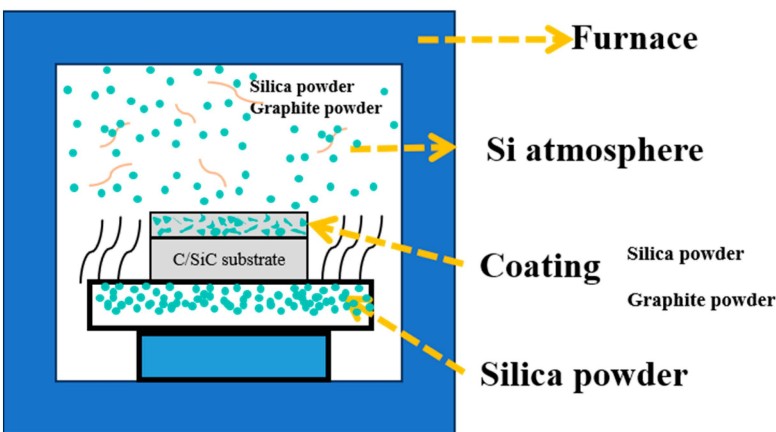

**Figure 1.** Schematic of the preparation of Si/SiC coatings.

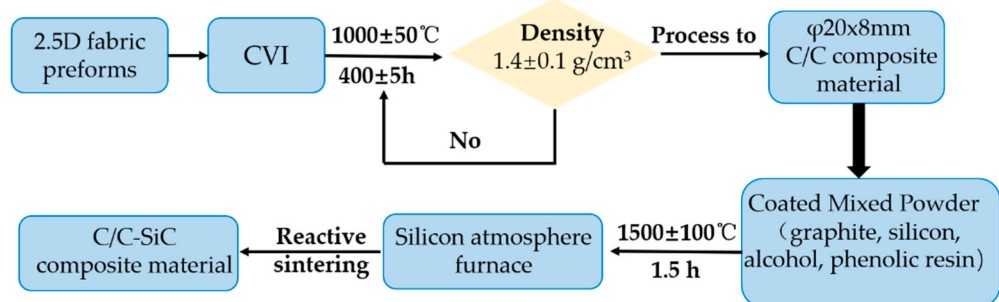

**Figure 2.** Schematic diagram of the preparation process of Si/SiC coatings on C/C-SiC composite.

**Table 1.** The density of the specimens.

| Sample Type | Density (g/cm³) | Reference |
|---|---|---|
| C/C-SiC | 2.0 ± 0.2 | This work |
| C/C | 1.4 ± 0.1 | This work |
| C/C-SiC | 2.0 | [32] |
| C/C-SiC | 2.0 | [33] |
| C/C-SiC | 2 | [34] |
| C/C-SiC | 1.848–1.921 | [10] |
| C/C | 1.03–1.47 | [10] |
| C/C | 1.5 | [35] |

*2.2. Characterization and Wear Test*

The Si/SiC coatings were prepared on C/C-SiC composites with a size of $\Phi$ 30 × 8 mm. The prepared friction wear specimens had a coating thickness of 500 ± 100 μm and a substrate thickness of 8 mm. The coatings were ground by a grinding machine to a certain surface roughness with the tested Ra value of about 0.4 before the wear test. The C/C-SiC composite material without a coating was ground to the same roughness for wear testing. The frictional coefficient and the wear mass loss of the samples were tested through a pin-on-disk friction and wear tester (HT1000, Lanzhou Zhongke Kaihua Technology Development Co., Ltd., Lanzhou, China). The wear test was conducted at room temperature with a load of 10 N, at a rotational speed of 251 mm/s, for a test time of 30 min. The N80 bulk was selected and cut into a cylindrical pin with a size of $\Phi$ 3 × 10 mm, serving as a counterpart in the friction and wear test. The stability of the wear process was analyzed by the frictional coefficient. The samples were weighed before and after the wear test through an electronic

balance with accuracy of 0.0001 g. Then, the wear mass losses were calculated. After the wear tests, the wear morphologies were characterized through SEM (VK-X3000, Keyence, Osaka, Japan). The phases were identified by X-ray diffraction (XRD-6000, Shimadzu, Kyoto, Japan) with Cu Kα radiation at a step of 0.02° and a scanning rate of 4°/min, with 2θ ranging from 20 to 80°.

## 3. Results and Discussion

### 3.1. Characterization of C/C-SiC Composites

Figure 3 shows the X-ray diffraction patterns of the C/C–SiC composites. The C/C–SiC composites were composed of C, Si and SiC phases. Five characterized diffraction peaks were observed at the 2θ angles of 35.597°, 41.383°, 59.977°, 71.777° and 75.492°, which could be attributed to the β-SiC phase according to PDF card No. 73-1663. Four diffraction peaks were observed at the 2θ angles of 28.442°, 47.302°, 56.121° and 76.377°, which could be attributed to the Si phase according to PDF card No. 75-0590. Additional diffraction peaks were observed at the 2θ angles of 26.603° and 44.669°, which were attributed to pyrolytic carbon (PyC) according to the PDF card No. 89-8487. The SiC, Si and C content in C/C–SiC composites was 71.6%, 11.4% and 17.0%, respectively, determined by the K value method of X-ray diffraction [27–29].

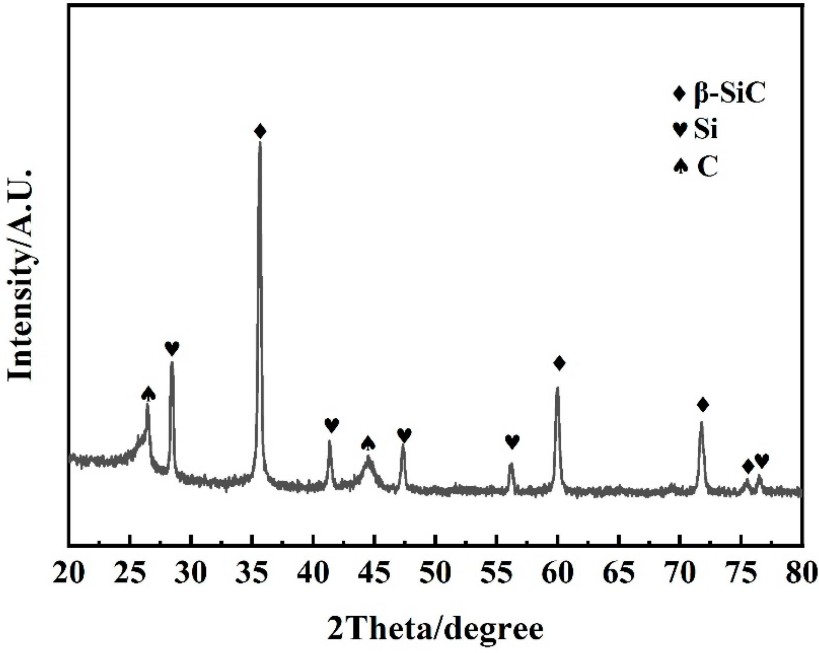

**Figure 3.** X-ray diffraction patterns of C/C–SiC composites.

Figure 4 shows the microstructure of the C/C-SiC composite material. Regions A and B reflect the short-cut fiber mesh region and the non-woven fiber cloth region. As shown in Figure 4a, the C/C-SiC composite material was mainly composed of SiC and PyC regions, with no obvious defects such as cracks and pores on its surface. The braking process occurred mainly on the surface of the composite material. Figure 4b shows the microstructure of a cross-section of the C/C-SiC composite material, which was composed of alternating short-cut fiber mesh areas and non-woven fiber cloth areas. There was a clear needle-punched area between area A and area B. The C/C-SiC composite material was a typical heterogeneous material. Magnified views of regions A and B are shown in Figure 4c,d. The short fiber network region was composed of SiC and PyC phases, and the pores outside PyC were filled with SiC, as shown in Figure 4c. The non-woven fiber fabric area had a "fiber bundle + PyC + SiC" structure, as shown in Figure 4d. The fiber bundles were relatively intact and had a smooth surface, indicating

that the PyC layer effectively protected the carbon fibers from reacting with liquid Si during high-temperature infiltration.

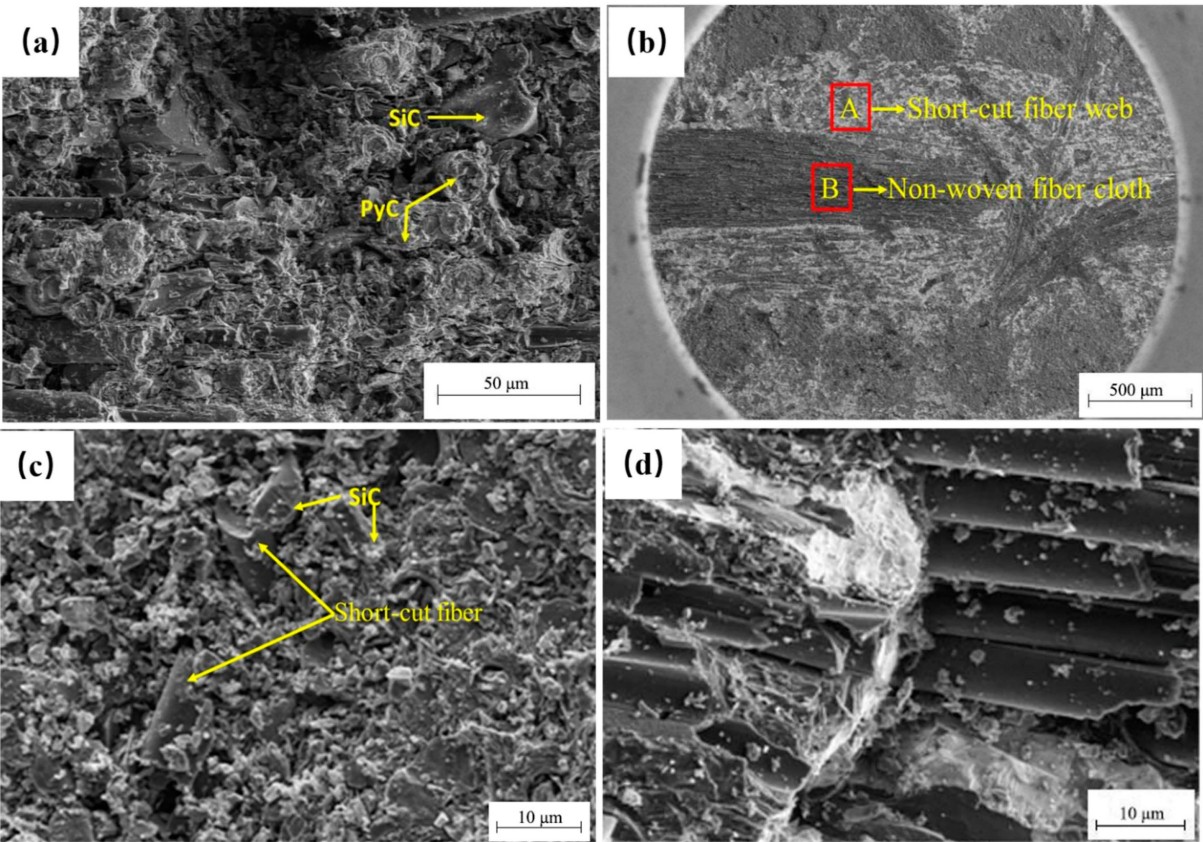

**Figure 4.** Microstructure of the surface and cross-section of C/C-SiC composite material: (**a**) surface morphology, (**b**) cross-section microstructure, (**c**) magnified microstructure of region A in (**b**), (**d**) magnified microstructure of region B in (**b**).

*3.2. Characterization of Si/SiC Coating*

Figure 5 shows the surface morphologies of SiC coatings with different carbon silicon ratios. It could be seen from the morphology that fine discontinuous SiC particles were formed during the coating preparation process when the weight ratio of carbon to silicon was 1:1, as shown in Figure 5a. When the weight ratio of carbon to silicon was increased to 1:2, the size of silicon carbide in the coating increased, as shown in Figure 5b. When the carbon to silicon ratio was 1:3, the silicon carbide particle size increased and became denser, as shown in Figure 5c, which could be helpful to enhance its wear performance. When there was an excess of silicon, the silicon carbide particles in the coating showed an irregular shape. There was a gap between the particles, as shown in Figure 5d. Therefore, as seen in the surface morphologies of the coatings, the coatings prepared with a carbon to silicon ratio of 1:3 were the densest ones.

Figure 6 shows the cross-sectional microstructures of Si/SiC coatings with different carbon silicon ratios. The thicknesses of the coatings ranged from 300 to 500 μm. The coating and substrate were tightly bonded. The Si/SiC phase in the coating grew together with the Si/SiC phase in the substrate. The coating had an alternating structure between the dark region (SiC) and the bright region (Si). As the carbon silicon ratio increased, the SiC gradually transitioned from a discontinuous state to a continuous state.

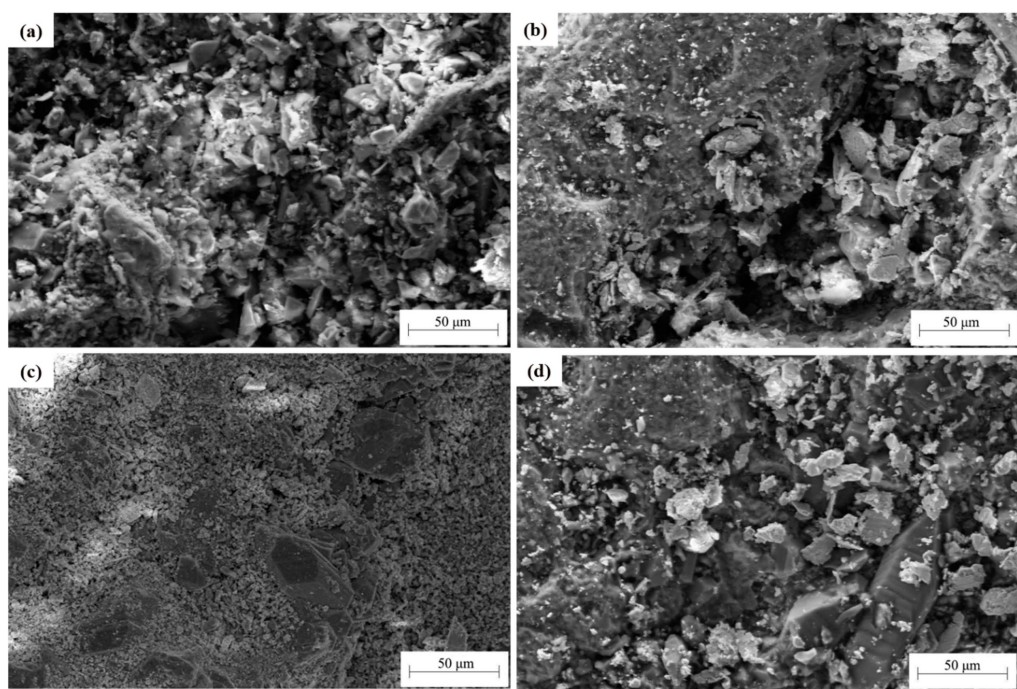

**Figure 5.** Surface morphologies of Si/SiC coatings with C to Si ratios of (**a**) 1:1, (**b**) 1:2, (**c**) 1:3, (**d**) 1:4.

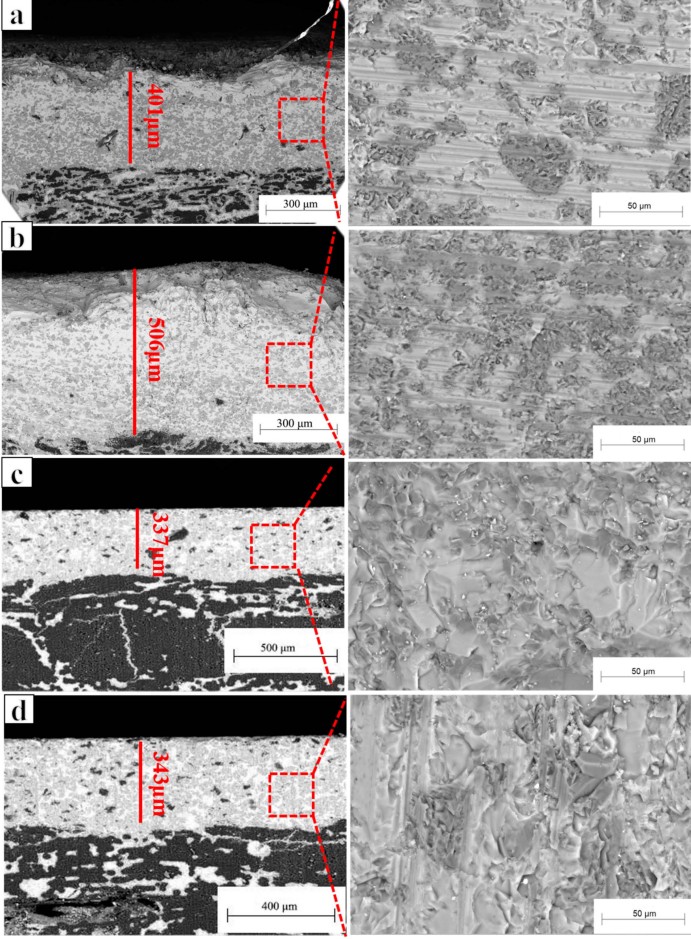

**Figure 6.** Cross-sectional microstructures of Si/SiC coatings with C to Si ratios of (**a**) 1:1, (**b**) 1:2, (**c**) 1:3, (**d**) 1:4.

### 3.3. X-ray Diffraction Patterns of Si/SiC Coatings

Figure 7 shows the X-ray diffraction patterns of the Si/SiC coatings on the C/C-SiC composites. The Si/SiC coatings were composed of Si and SiC phases. The SiC and Si content in the Si/SiC coating is shown in Table 2, as determined by the K value method of X-ray diffraction. Among the four coatings, the coating with a carbon to silicon ratio of 1:1 had the lowest SiC content of about 87.1%. From Equation (1), it can be seen that when the carbon and silicon reacted completely to form silicon carbide, the carbon/silicon atomic ratio was 1:1, and the mass ratio of carbon and silicon was 12:28, about 1:2.3 in atomic ratio. When the carbon silicon mass ratio was 1:1, its atomic ratio was much smaller than 1:1, and the silicon carbon in the coating slurry did not fully react, resulting in lower silicon carbide content in the coating. However, due to the melting and volatilization of silicon atoms at high temperatures, an excess of silicon atoms in the slurry was needed in the actual reaction to allow the carbon atoms in the slurry to react completely to form silicon carbide. When the carbon to silicon ratio reached 1:3, the maximum content of silicon carbide was 93%. The atomic ratio of carbon to silicon was close to 1:1, and the silicon carbon reaction was complete. In the XRD patterns, with a carbon silicon ratio of 1:3 and 1:4, diffraction peaks were observed at positions 34.116° and 34.734°, matched with the SiC (101) and (015) crystal planes according to PDF card No. 73-1663 and PDF card No. 89-2219. The appearance of low-index crystal planes indicated the tighter arrangement of SiC atoms in the coating, resulting in the higher density of the coating. When the carbon to silicon ratio was 1:4, the silicon content slightly increased due to the excess of silicon powder. The volume shrinkage caused by the melting of silicon caused a decrease in the crystallinity of silicon carbide.

$$Si(s) + C(s) = SiC(s) \tag{1}$$

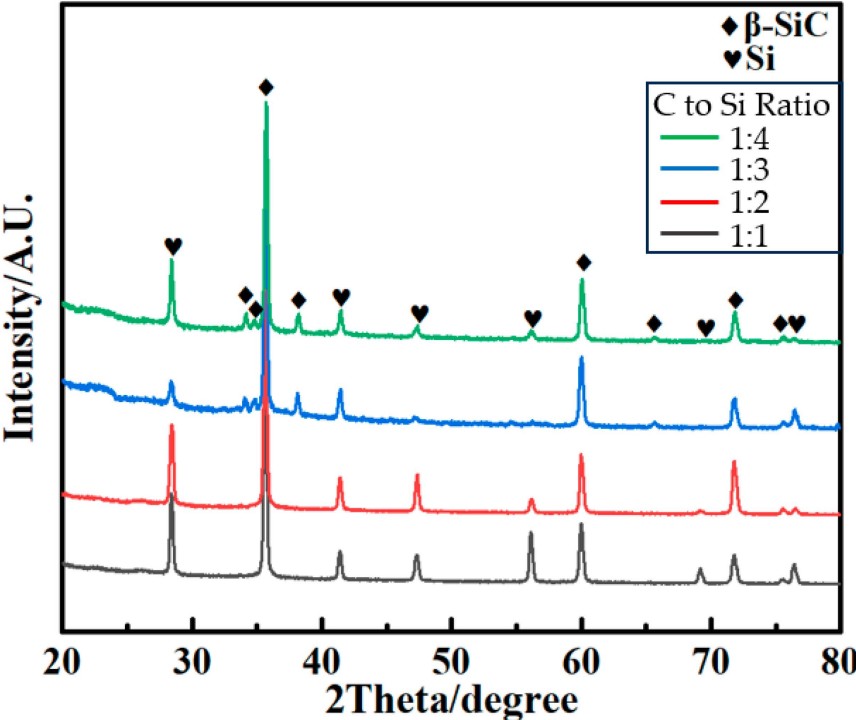

**Figure 7.** X-ray diffraction patterns of different C/Si ratio coatings.

**Table 2.** Calculated content of Si/SiC coatings.

| C to Si Ratio (wt.) / Phase Content (wt.%) | SiC | Si |
|---|---|---|
| 1:1 | 87.1 | 12.9 |
| 1:2 | 87.6 | 12.4 |
| 1:3 | 93.0 | 7.0 |
| 1:4 | 90.3 | 9.7 |

There were two diffraction peaks in the beta SiC peaks beyond 70 degrees, located at 71.769 degrees and 75.500 degrees, respectively, which corresponded to the (311) and (222) crystal planes. The diffraction peak of the (222) crystal plane varied very little with the carbon to silicon ratio, indicating that β-SiC did not have a clear (222) preferred orientation. When the carbon to silicon ratio was 1:1, the carbon atom concentration was low, which could not support the continuous nucleation and growth of β-SiC. The crystallinity of β-SiC was weak, so the intensity of the (311) crystal plane diffraction peak in the XRD pattern was weak. When the carbon to silicon ratio was 1:2, the (311) crystal plane diffraction peak was enhanced to a certain extent, indicating that, at this time, the preferred (311) orientation of β-SiC was gradually increasing. When the carbon to silicon ratio was 1:3, the reaction between silicon and carbon was complete. At this time, the diffraction peak intensity of the (311) crystal plane was weakened, but the diffraction peak intensities of the (220) and (111) crystal planes were enhanced, indicating that, at this time, the preferred (111) and (220) orientations of β-SiC were gradually increasing, suppressing the preferred (311) orientation. When the carbon to silicon ratio was 1:4, there was a severe excess of silicon atoms, which inhibited the growth of silicon carbide particles, and there were no significant changes in the (311) diffraction peak.

*3.4. Tribological Properties*

3.4.1. Friction and Wear Behavior

The frictional coefficients varied as a function of the wear time. Figure 8 shows the frictional coefficients of the Si/SiC coatings and the C/C-SiC substrate. Usually, the frictional wear contains three segments, including primary abrasion, normal abrasion and rapid abrasion. As seen in Figure 8a, only primary abrasion and normal abrasion occurred during the testing. The C/C-SiC composite substrate had the lowest frictional coefficient of about 0.68 at the initial period of the friction test. As the wear time increased, the frictional coefficient curve ascended gradually to a plateau of about 0.63.

The decrease in the frictional coefficient as compared with the coatings during the normal abrasion stage could be attributed to the lubrication of the free graphite phase in the matrix surface. The carbon fibers in the C/C-SiC composite substrate had a certain self-lubricating ability, which was one of the reasons for its lower frictional coefficient. The carbon fibers had a high degree of crystallinity and a parallel-aligned crystal structure, with a smooth surface and low surface energy. In the friction process, the contacts between carbon fibers would release carbon on the surfaces of the carbon fibers, forming a very thin layer of carbon film, which would play a self-lubricating role.

Figure 8b and Table 3 show the statistic frictional coefficients of the substrate and coatings with different ratios of carbon to silicon. Among the four coatings, the coating with a carbon to silicon ratio of 1:1 had the lowest frictional coefficient of about 0.7. Due to insufficient silica powder, it was not possible to grow a dense SiC ceramic layer. Meanwhile, excessive carbon powder would form a lubricating phase. With the increase in silicon content, the frictional coefficient of the coating increased continuously. When the ratio of carbon to silicon powder was 1:3, the frictional coefficient of the coating was the highest. At this ratio, the carbon reacted with the silicon powder fully to generate a relatively dense SiC coating in situ. The coating could not be lubricated by the excess carbon powder. As compared to the 1:2 coating, the 1:3 coating was relatively dense. Therefore, the frictional coefficient was increased significantly. When the ratio of carbon to silicon powder was

1:4, the frictional coefficient was reduced to 0.84. This was attributed to the fact that the production of silicon carbide through the in situ reaction was accompanied by volume shrinkage and produced microcracks. The excessive silicon powder filled the cracks and gaps and entered the lubrication phase, resulting in a decrease in the frictional coefficient. Therefore, as a brake disc wear-resistant material, the coating prepared with a carbon to silicon ratio of 1:3 could significantly improve the frictional coefficient of the material and improve the braking effect of the brake disc.

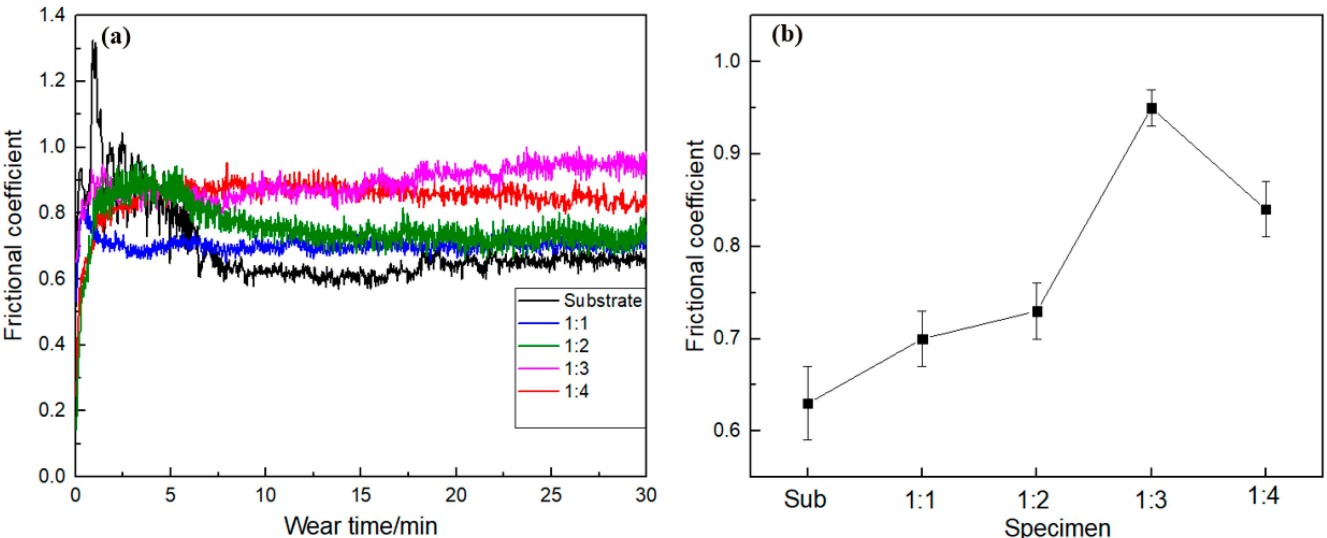

**Figure 8.** Frictional coefficients of Si/SiC coatings and the substrate. (**a**) Frictional coefficient curve. (**b**) Average value of frictional coefficient.

**Table 3.** Statistic frictional coefficients of the SiC coatings and the substrate.

| Type | Sub | Coating | | | |
|---|---|---|---|---|---|
| | | 1:1 | 1:2 | 1:3 | 1:4 |
| Frictional coefficient | 0.63 ± 0.04 | 0.70 ± 0.03 | 0.73 ± 0.03 | 0.95 ± 0.02 | 0.84 ± 0.03 |

Figure 9 shows the wear rates of the coatings, C/C-SiC composite substrate and corresponding counterparts after the wear tests. The coating's wear rate decreased with the increase in the Si content. The Si/SiC coating prepared at the carbon to silicon ratio of 1:3 exhibited the lowest wear rate of $3.2 \times 10^{-3}$ mg·N$^{-1}$·m$^{-1}$ among the four coatings. Moreover, the wear rate of the counterpart was greater than that of the SiC coatings, but much lower than that of the C/C-SiC composite substrate. As compared to the wear rate of the C/C-SiC composite substrate, with the value of $16.3 \times 10^{-3}$ mg·N$^{-1}$·m$^{-1}$, the coatings' wear rates were significantly reduced, which reflected that the coatings' wear resistance was higher than that of the substrate. The main reason for the high wear rate of the substrate was the low hardness and high brittleness of the carbon ceramic substrate; it tended to be abrased and peeled off, resulting in a high wear rate. The SiC coating could effectively reduce the amount of wear to protect the C/C-SiC composite substrate. However, the amount of wear mass loss of the counterpart was slightly greater than that of the coating. The N80 counterpart's hardness was about 60–62 HRC, which was approximately equal to 700–760 HV. Meanwhile, the microhardness of the SiC coating was above 2000 HV. The SiC coating was thus much harder than the counterpart. Due to the silicon carbide coating and the lubrication of graphite powder and silicon powder, the wear rate of the coating and the corresponding counterpart was significantly reduced. Therefore, combined with both the wear rate and frictional coefficient, the SiC coating prepared at the carbon to silicon

ratio of 1:3 is preferred for its optimized wear resistance and high frictional coefficient and wear rate.

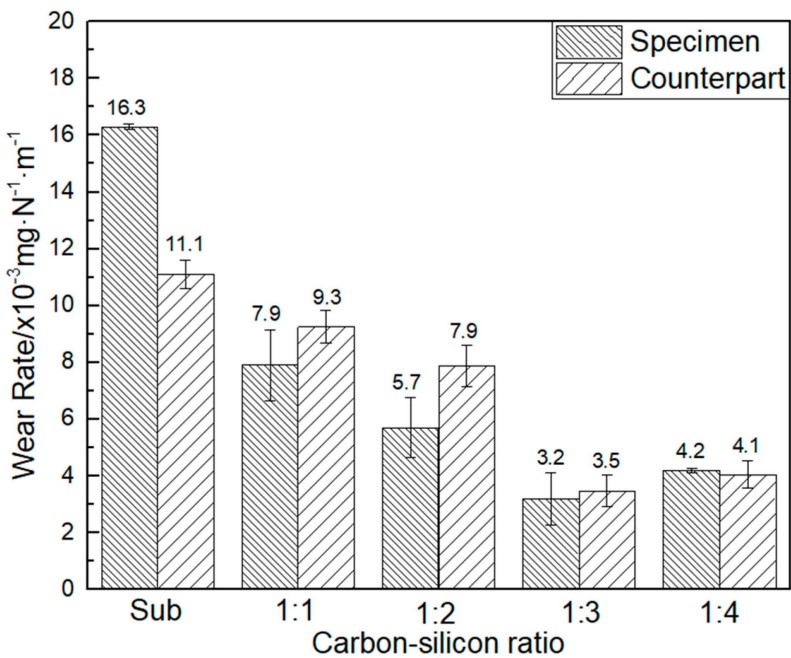

**Figure 9.** Wear rates of the specimen and counterparts.

### 3.4.2. Worn Surface of the Coatings

The worn surface morphologies of the C/C-SiC composite substrate and SiC coatings prepared at the carbon to silicon ratios of 1:1, 1:2, 1:3 and 1:4 are shown in Figure 10. There was some wear debris adhering to the tracks, and some typical plough zones were observed on the worn surface, which implied that the deformation and fracture of asperities appeared due to the extrusion of particles during wear. Therefore, the coating's wear mechanism was abrasive wear at room temperature. Combined with the EDS analysis (Table 4), it was clearly observed that the wear track was covered by a thick oxidized debris layer. It is worth noting that the Ni element was detected in the debris, which meant that most debris peeled away from the counterparts (N80). Moreover, when the carbon to silicon ratio was 1:3, the wear debris on the coating's surface was significantly smaller than that of other coatings, as seen in Figure 10c. The dense silicon carbide coating effectively protected the substrate and reduced the wear mass loss. Figure 10e shows the worn surface morphologies of the substrate. There were large amounts of wear debris on the worn surface of the C/C-SiC composite substrate. The great plastic deformation as well as surface peeling in the counterpart caused by the hard SiC formed the debris in the friction process. Since the substrate was not protected by a dense silicon carbide coating, the wear debris could lead to severe three-body abrasive wear in the sliding processing, which could aggravate the wear process and produce more grooves on the surface of the substrate, finally increasing the wear mass loss. Therefore, the coating could effectively protect the substrate and reduce the wear behavior. When the carbon to silicon ratio was 1:3, the coating had the lowest wear rate.

**Table 4.** Point analysis of elements in the SiC coating prepared at carbon to silicon ratio of 1:2.

| Content (wt.%) | C | O | Si | Cr | Ni |
|---|---|---|---|---|---|
| Point 1 | 18.52 | 7.97 | 5.95 | 14.20 | 53.35 |
| Point 2 | 34.53 | 4.24 | 61.23 | - | - |

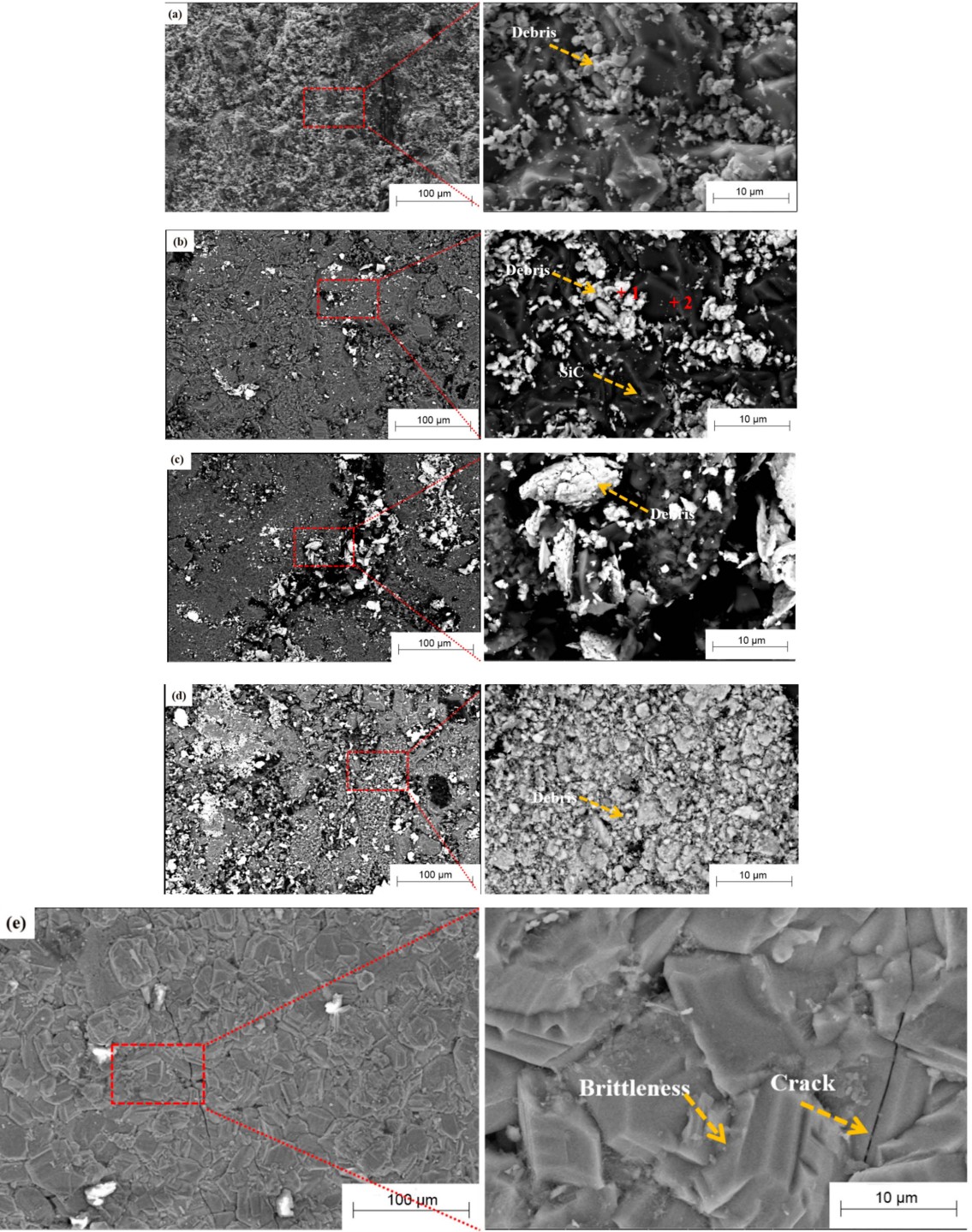

**Figure 10.** Worn surface morphologies of the coatings and substrate: (**a**) 1:1, (**b**) 1:2, (**c**) 1:3, (**d**) 1:4 and (**e**) substrate.

The worn surface morphologies of the wear debris are shown in Figure 11. The particle size of the debris was about 50 µm–100 µm. The debris had an irregular morphology, such as flakes and blocks. Through the analysis of the EDS, as shown in Table 5, there were flaked SiC particles in the abrasive chips at point 1 and there were also elements of Cr and Ni in addition to the elements of Si, C and O at point 2, which indicated that, during the abrasion process, the N80 pins were ploughed, and the metal materials were embedded in the rigid SiC particles. Meanwhile, during the abrasion process, the SiC particles were flaked and embedded in the N80 pins as well.

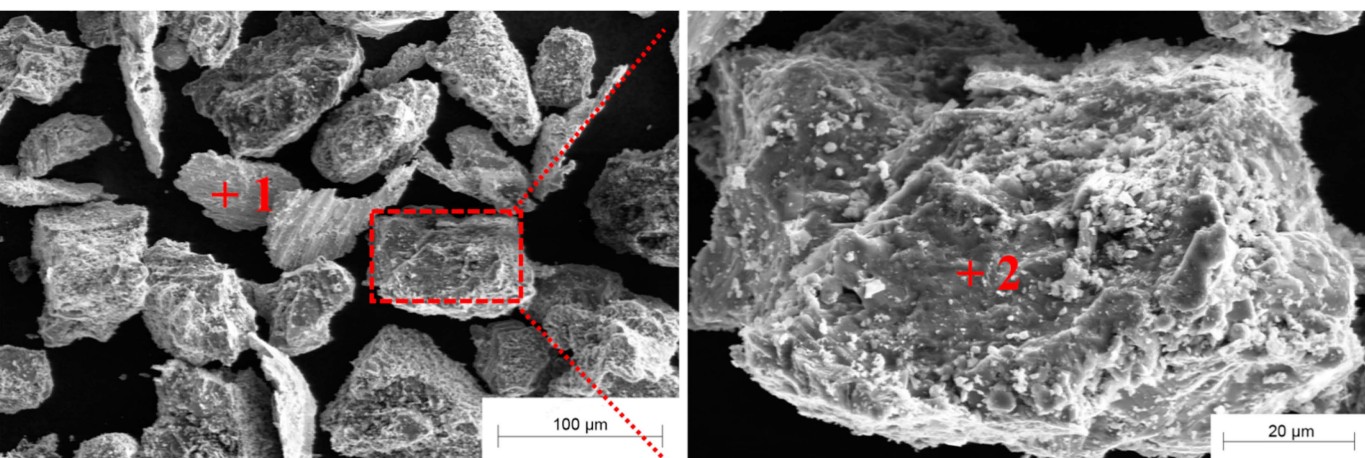

**Figure 11.** Worn surface morphologies of the wear debris.

**Table 5.** Point analysis of elements in the wear debris.

| Content (wt.%) | C | O | Si | Cr | Ni |
|---|---|---|---|---|---|
| Point 1 | 28.57 | - | 71.43 | - | - |
| Point 2 | 51.25 | 6.23 | 30.9 | 2.67 | 9.75 |

3.4.3. Worn Surface Analysis of the Counterparts

Figure 12 shows the worn surface morphologies of the corresponding counterparts. Due to the difference in hardness between the coating and the counterpart, the counterparts' worn surfaces showed more micro-cutting furrows and plowings than the coatings. As seen in Figure 12a, the plowing grooves on the worn surfaces of the counterpart corresponding to the coating prepared at the carbon to silicon ratio 1:1 was more serious as compared with the others. This phenomenon was related to the low silicon content of the coating; the silicon carbide formed in the coating peeled off during the wear and aggravated the wear process. With the increase in the silicon content, the plowing grooves on the counterpart's surface were reduced, as seen in Figure 12b. When the ratio of carbon to silicon was 1:3, the number of the plowing grooves on the counterpart's surface was much smaller as compared to the others, as seen in Figure 12c. As the coating was relatively dense, few SiC particles were removed and embedded into the N80 counterpart. Meanwhile, it had the lowest wear rate of $3.5 \times 10^{-3}$ mg·N$^{-1}$·m$^{-1}$ among the four counterparts. When the carbon to silicon ratio reached 1:4, the residual fine silicon carbide particles were embedded in the counterpart, as seen in Figure 12d.

Figure 13 shows the element mapping distribution on the N80 counterpart's worn surface. The counterpart's worn surface showed obvious elemental segregation. The Si and C elements were enriched here, which indicated that the counterpart's surface contained SiC particles. Table 6 shows the point analysis of elements in the N80 counterpart after the friction and wear tests. As shown in Figure 14, four points were selected for analysis. It was found that Ni, Cr, Si and C were detected in the compositional analysis of all four points, which indicated that abrasive and adhesive wear occurred on the worn surface of the N80 pin during the wear test. The content of Si and C in the black hard particles was significantly higher than that in the N80 matrix region. This suggested that the SiC hard particles flaking off from the coating against the N80 pins were tightly bound together in the grooves and pits on the worn surfaces due to shear stress, compressive stress and frictional heat. Therefore, the coating could not only improve the wear resistance of the substrate but also reduce the wear of the counterpart.

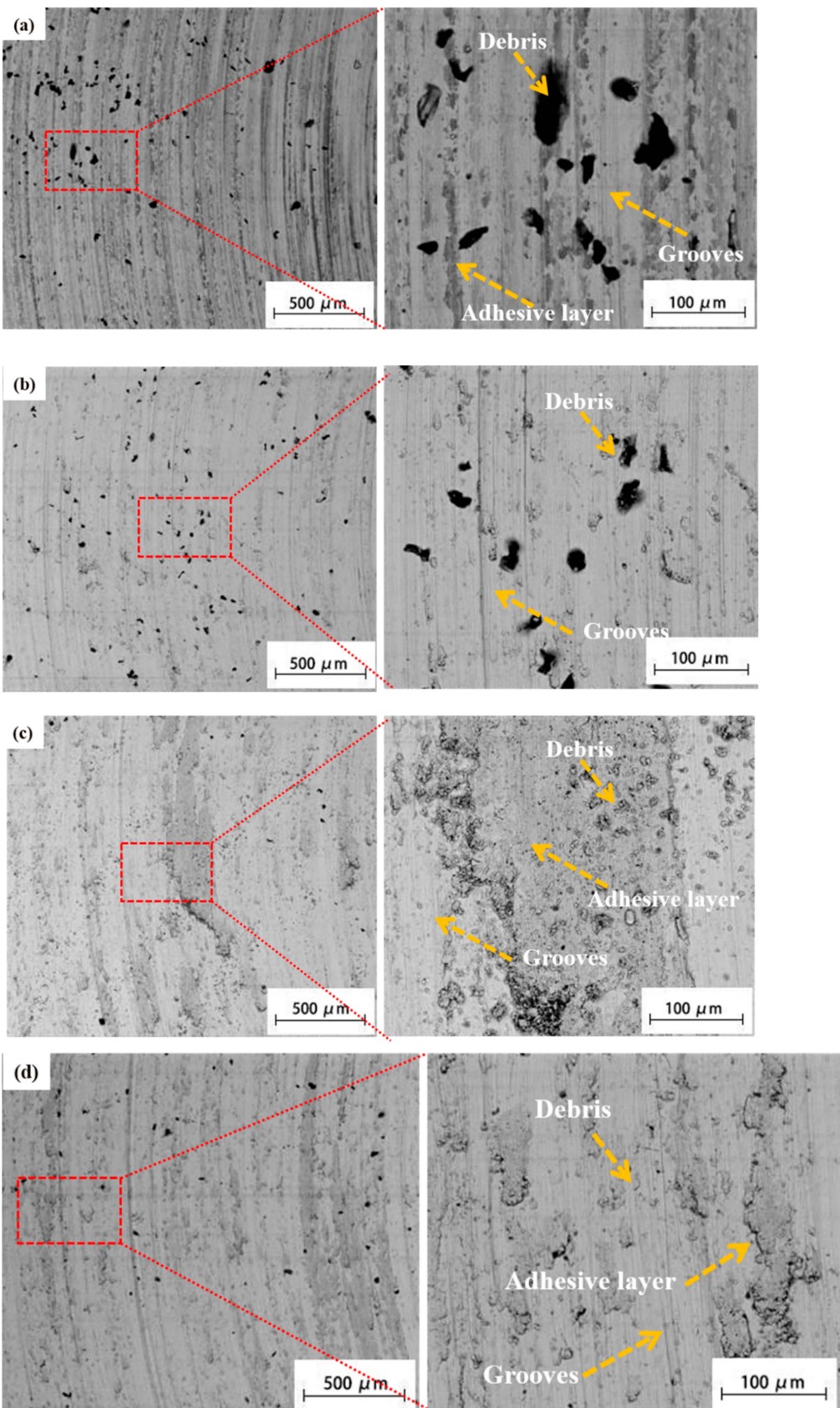

**Figure 12.** Worn surface morphologies of the corresponding counterparts: (**a**) 1:1, (**b**) 1:2, (**c**) 1:3, (**d**) 1:4.

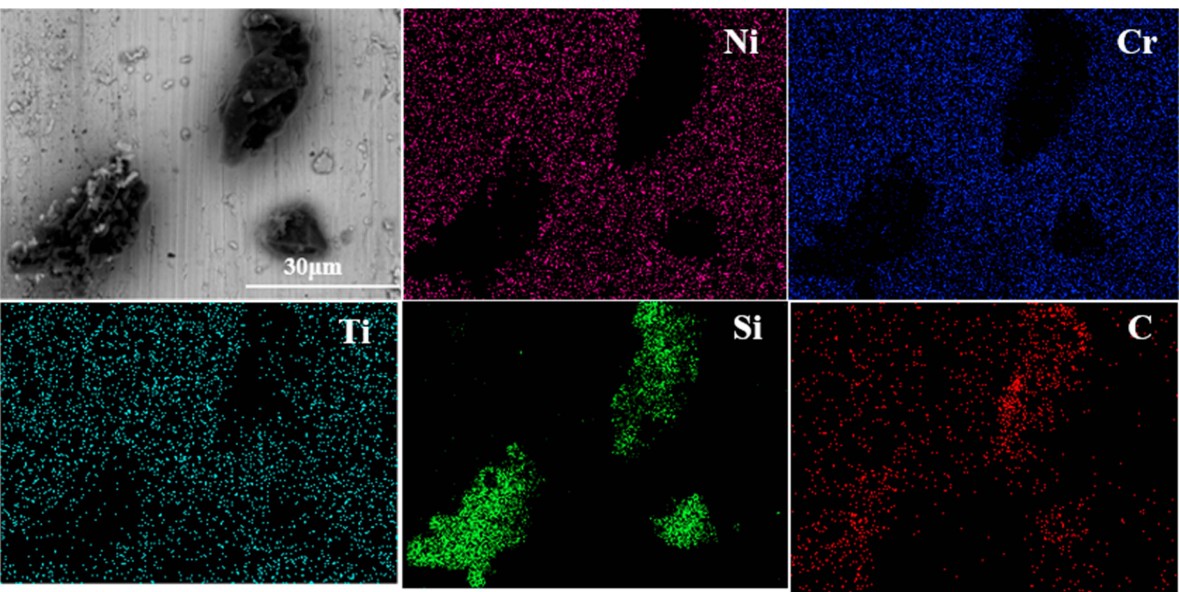

**Figure 13.** Cross-sectional distribution of elements in the N80 counterpart.

**Table 6.** Point analysis of elements in the counterpart.

| Content (wt.%) | Ni | Cr | Ti | Al | Si | C | O |
|---|---|---|---|---|---|---|---|
| Point 1 | 75.52 | 19.10 | 2.20 | - | 3.09 | 0.09 | - |
| Point 2 | 3.50 | 1.32 | - | 1.17 | 77.58 | 11.91 | 4.51 |
| Point 3 | 2.31 | 1.05 | - | - | 91.19 | 5.45 | - |
| Point 4 | 77.94 | 16.39 | 1.41 | - | 4.13 | 0.13 | - |

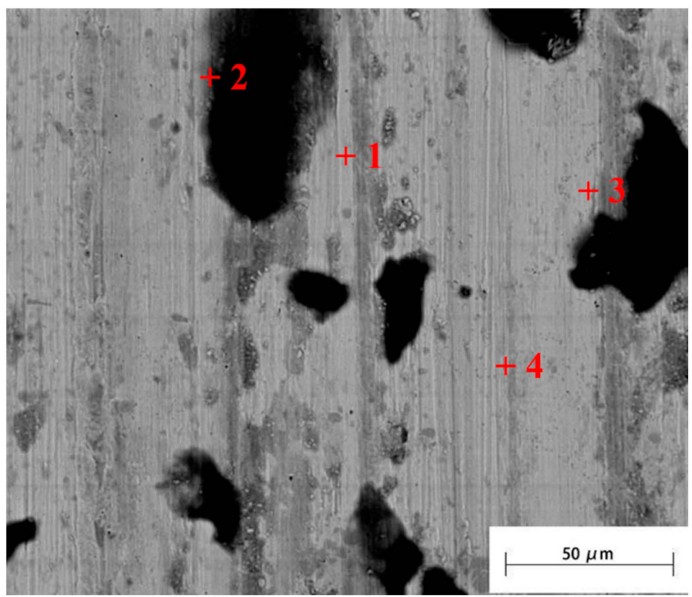

**Figure 14.** Point analysis of elements in the N80 counterpart.

Figure 15 shows the wear schematic of the coatings with different carbon to silicon ratios. The coating's wear resistance was closely related to the microstructure and density of the coating. When the ratio of graphite powder to Si powder was 1:1, the graphite powder reacted with the Si powder to form SiC, resulting in volume shrinkage. However, the remaining graphite and the matrix reacted with liquid Si to form SiC due to the siphon

effect, resulting in volume expansion, the non-bonding of SiC, discontinuous coating pieces, and lubrication, so the friction coefficient was the lowest. However, the non-dense SiC coating increased the amount of wear as small silicon carbide particles peeled off or were embedded into the counterpart during the wear process.

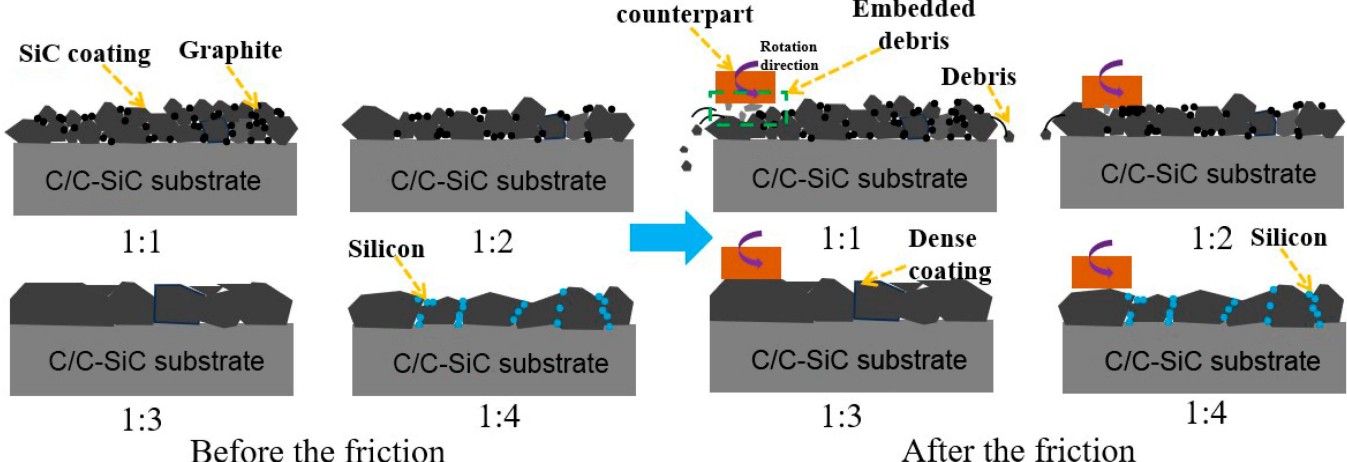

**Figure 15.** Wear schematic of coatings with different carbon to silicon ratios.

When the ratio of graphite powder to Si powder was 1:2, accompanied by the increase in Si powder content, the SiC generated by the in situ reaction between graphite powder and Si powder increased, resulting in a increase in the SiC content of the coating. It finally resulted in reduced wear debris and reduced wear mass loss. When the ratio of graphite powder to Si powder was increased to 1:3, the graphite powder and Si powder reacted in situ to form a relatively continuous SiC layer, and the coating was relatively dense. Therefore, such content increased the frictional coefficient of the coating and reduced the wear mass loss simultaneously.

Moreover, when the ratio of graphite powder to Si powder was 1:4, the Si powder was excessive. The gap in the SiC layer was filled with residual silicon powder to enter the lubricating phase, so the frictional coefficient decreased.

Therefore, when the ratio of graphite powder to Si powder was 1:3, both the high hardness of SiC and the dense coating contributed simultaneously to enhancing the coating's wear resistance.

## 4. Conclusions

Si/SiC coatings were prepared by a combination of chemical vapor infiltration and reactive sintering to improve the wear resistance of C/C-SiC composites. The influence of the carbon to silicon ratio and phase composition on the microstructure and wear resistance of Si/SiC coatings was investigated. The conclusions can be summarized as follows.

1. The C/C-SiC composite material was mainly composed of the alternate stacking of a weft-free fabric and short fiber mesh with three phases of C, Si and SiC.
2. With the increase in SiC content, the coating's frictional coefficient increased and the wear rate decreased.
3. When the carbon to silicon ratio in the slurry was 1:3, the Si/SiC coating had 93.0% SiC, which was the highest content of SiC, as well as the highest frictional coefficient of 0.95 and the lowest wear rate of $3.2 \times 10^{-3}$ mg·N$^{-1}$·m$^{-1}$.
4. The Si/SiC coating exhibited abrasion, adhesion and oxidation wear on N80 friction pairs.

In the future, Si/SiC coatings will be used for moving parts with long-life C/C or C/C-SiC composites. Meanwhile, the processing will be optimized and the costs will be reduced further.

**Author Contributions:** Conceptualization, D.Z., P.G. and K.C.; methodology, P.G., D.Z., K.C., Q.G. and H.C.; software, P.G., D.Z. and K.C.; validation, P.G., B.C., Q.G. and A.N.; formal analysis, P.G., D.Z. and K.C.; investigation, P.G., K.C., Q.G., Q.L. and W.K.; data curation, Q.G.; writing—original draft preparation, D.Z.; writing—review and editing, D.Z. and P.G.; project administration, Q.G.; funding acquisition, P.G. and Q.G. All authors have read and agreed to the published version of the manuscript.

**Funding:** This work was funded by the National Natural Science Foundation of China (51771140); the Foreign Experts Program of the Ministry of Science and Technology of China (G2022040016L); the Youth Innovation Team of Shaanxi Universities: Metal Corrosion Protection and Surface Engineering Technology, Research and application of key component materials for engines, Shaanxi Provincial Natural Science Foundation (2023-JC-YB-380); the Shaanxi Provincial Key Research and Development Project (2019ZDLGY05-09); and the Xi'an Science and Technology Plan Project (23LLRHZDZX0019).

**Institutional Review Board Statement:** Not applicable.

**Informed Consent Statement:** Not applicable.

**Data Availability Statement:** Data are contained within the article.

**Conflicts of Interest:** Daming Zhao, Kaifeng Cheng, Hao Cheng, Qiao Li and Wenjie Kang were employed by the company. The remaining authors declare that the research was conducted in the absence of any commercial or financial relationships that could be construed as a potential conflict of interest.

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
