# Peer review of "Investigation of Impact of C/Si Ratio on the Friction and Wear Behavior of Si/SiC Coatings Prepared on C/C-SiC Composites by Slurry Reaction Sintering and Chemical Vapor Infiltration"

_coatings, doi:10.3390/coatings14010108_

Round 1

Reviewer 1 Report

Comments and Suggestions for Authors

The manuscript titled "Investigation of Friction and Wear Behavior of Si/SiC Coatings Prepared on C/C-SiC Composites by Slurry Reaction Sintering and Chemical Vapor Infiltration" needs attention in the following aspects:

1. The title of the manuscript contains too many condensed terms. Kindly modify the title suitably.

2. The abstract needs to be included briefly with the gap identified and the need for the current study. 

3. The novelty of the current work has not been portrayed in the introduction section. Authors may refer to some of the following recently published articles:

10.1016/j.jmrt.2021.09.090, 10.1016/j.jeurceramsoc.2019.09.046, 10.1016/j.corsci.2021.109331

4. All the condensed forms are to be expanded during their first usage wherever applicable. Kindly avoid condensed forms in the abstract and conclusion sections.

5. In Fig. 3a, how SiC and PyC were demarcated? Kindly justify.

6. Kindly discuss about the variation in beta-SiC peaks beyond 70 degrees for each composition in Fig. 6.

7. Reasons for the drastic increase in friction coefficient between 1:2 and 1:3 is missing. Kindly include.

8. What is the composition of wear debris? Was it analyzed? Kindly provide the data.

9. Conclusion section has to be elaborated further by including the limitations and future scope of the experimental work.

Comments on the Quality of English Language

Grammar and language need substantial revision throughout.

Reviewer 2 Report

Comments and Suggestions for Authors

The work by Zhao et al. (Investigation of Friction and Wear Behavior of Si/SiC Coatings Prepared on C/C-SiC Composites by Slurry Reaction Sintering and Chemical Vapor Infiltration) reports the employment of  Si/SiC coatings to improve wear resistance of C/C-SiC composite matrix. For this, they used a combination of chemical vapor infiltration and reactive sintering. The authors observed that at the optimal condition of the carbon-silicon ratio in the slurry, they could obtain an ideal wear resistance.    

The research seems to be interesting and provides valuable information to this research field. However, I noticed some major issues that must be clarified. Therefore, I recommend the publication of the report, after these issues are addressed.

The last paragraph of the Introduction must be extended. The authors must discuss the other studies in the literature to improve the wear resistance of C/C-SiC and provide a contribution of the manuscript to this research field. Also, the authors must indicate the content of the research and some significant results in the paragraph.

Please reconsider the Material Preparation section. It is very hard to follow. A scheme showing the all preparation would be helpful.

Please delete Si/SiC coating from the heading of section 3.1. I think all headings must be reconsidered.

Line 139, please fix Fig1a.

Line 141, please fix Fig1b.

Line 143, please fix Fig1c.

Line 145, please fix Fig1d.

I think the authors must provide more cross-section SEM images depicted in Figure 5 at higher magnification. The higher-magnified images would provide valuable insight for the evaluation. Also, show the film thickness on the images.

Please delete the % for the C to Si ratio in Table 1. 

Comments on the Quality of English Language

 Moderate editing of English language required

Reviewer 3 Report

Comments and Suggestions for Authors

The current paper reports the investigation of friction and wear behavior of Si/SiC coatings prepared on C/C-SiC composites by slurry reaction sintering and chemical vapor infiltration. The major criticism of this paper is that, there is a lack in proper literature review together with the novelty of the present work. The authors need to address these areas, without which, it will not be possible to accept the paper. Based on that, I recommend major revision of the manuscript. The specific comments are as follows:

1.     The title must be revised to be concise that reflect the content of the manuscript. In current form, it is not focused at all and all over the place.

2.     The abstract must be re-written in a focused way. The abstract should contain the main essence of the work including major findings with numerical data/values.

3.     The literature review is very weak and seems half-done! I am missing the critical analysis of the previously reported work in this area.

4.     The novelty of the present work compared to what has reported in the literature is missing.

5.     The aim/objective of the present work is missing which usually comes as a last paragraph of the introduction section.

6.     Detail characterization of the fabric preforms must be included in terms of optical/SEM images.

7.     The density of the specimens should be reported in tabular form together with in comparison with literature.

8.     Line 86: What was the roughness after grinding the specimens before wear testing?

9.     Line 91: The speed value should be given in mm/s.

10.  Line 194: It does not make any sense! What is frictional coefficient vibration?

11.  The evolution of the frictional coefficient for first few minutes (< 3 min) is questionable and most probably related to the roughness effect after grinding before the wear tests. This should be verified and reported/explained accordingly.

12.  Fig. 8: instead of mass loss, proper wear rate should be reported for better clarification of the results.

Comments on the Quality of English Language

 Minor editing of English language required

Round 2

Reviewer 1 Report

Comments and Suggestions for Authors

All the comments given by the reviewers have been addressed in the revised manuscript. The manuscript can now be accepted in its current form. Best wishes to all the authors.

Reviewer 2 Report

Comments and Suggestions for Authors

The authors improved the quality of the report after. I recommend the publication of the report. 

Reviewer 3 Report

Comments and Suggestions for Authors

Accept in present form

Comments on the Quality of English Language

Minor editing of English language required